# Whole-Genome Inter-Sex Variation in Russian Sturgeon (*Acipenser gueldenstaedtii*)

**DOI:** 10.3390/ijms23169469

**Published:** 2022-08-22

**Authors:** Gad Degani, Michal Nevo Sarel, Akram Hajouj, Avshalom Hurvitz, Isana Veksler-Lublinsky, Ari Meerson

**Affiliations:** 1MIGAL—Galilee Research Institute, P.O. Box 831, Kiryat Shmona 1101602, Israel; 2Faculty of Sciences, Tel-Hai Academic College, Upper Galilee 1220800, Israel; 3Department of Software and Information Systems Engineering, Faculty of Engineering Sciences, Ben Gurion University of the Negev, P.O. Box 653, Beer Sheva 8410501, Israel; 4Caviar Galilee Fishery, Kibbutz Dan 1224500, Israel

**Keywords:** sturgeon, *Acipenser*, genomics, variation, sex-based, aquaculture, endangered species

## Abstract

The Russian sturgeon (*Acipenser gueldenstaedtii*, AG) is an endangered fish species increasingly raised on fish farms for black caviar. Understanding the process of sex determination in AG is, therefore, of scientific and commercial importance. AG lacks sexual dimorphism until sexual maturation and has a predominantly octoploid genome without a definite sex chromosome. A conserved short female-specific genomic sequence was recently described, leading to the development of a genetic sex marker. However, no biological function has been reported for this sequence. Thus, the mechanism of sex determination and the overall inter-sex genomic variation in AG are still unknown. To comprehensively analyze the inter-sex genomic variation and assess the overall inter-species variation between AG and *A. ruthenus* (AR, sterlet), a related tetraploid sturgeon species, we performed whole-genome sequencing on DNA from 10 fish-farm-raised adult AG (5 males and 5 females). We produced a partially assembled, ~2390 MBp draft genome for AG. We validated in AG the female-specific region previously described in AR. We identified ~2.8 million loci (SNP/indels) varying between the species, but only ~7400 sex-associated loci in AG. We mapped the sex-associated AG loci to the AR genome and identified 15 peaks of sex-associated variation (10 kb segments with 30 or more sex-associated variants), 1 of which matched the previously reported sex-variable region. Finally, we identified 14 known and predicted genes in proximity to these peaks. Our analysis suggests that one or more of these genes may have functional roles in sex determination and/or sexual differentiation in sturgeons. Further functional studies are required to elucidate these roles.

## 1. Introduction

The Russian sturgeon (*Acipenser gueldenstaedtii*, AG), which produces black caviar, has decreased dramatically in natural habitats (Black Sea, Azov Sea, and Caspian Sea), which has significantly increased the market value of caviar [1]. This species belongs to Acipenseridae (common name: sturgeon), a primitive family of vertebrates whose origins date back to over 200 million years ago, and represents an interesting position in evolution for studying the biology of fish [2]. The females have higher economic value than the males, and sex determination and sexual differentiation of sturgeon are important for the aquaculture of this group. However, there is no morphological differences between the sexes, and until recently, the methods of distinguishing between males and females were complex and feasible only after sexual differentiation [3,4]. Moreover, there are no sex chromosomes, and some species belonging to Acipenseriformes, including AG, are octoploid species with 250 ± 8 chromosomes [5]. Several studies suggested a ZZ/ZW model of sex determination in sturgeon [6,7]. Several studies have been carried out in search of molecular sex markers in sturgeon species [8,9,10].

The genetic variability of seven sturgeon species was studied using mitochondrial DNA cytochrome oxidase subunit I (COI) and nine microsatellite markers [11]. Other studies measured transcriptomic differences between the gonads of males and females [9,10,12,13]. Recently, a conserved female-specific region was identified in six species of sturgeon, and a PCR-based genotyping method (AllWSex2) was developed [14]. The utility of this assay was further validated in studies of lake sturgeon (*A. fulvescens*) [15], and a modified protocol was tested and implemented in commercial aquaculture of Russian sturgeon [16]. While these findings have led to significant progress in practical applications, no biological function has been reported for this female-specific sequence. Furthermore, to our knowledge, no prior study has characterized the genome-wide genomic variation between the sexes in sturgeon. Thus, the mechanism of sex determination and the overall inter-sex genomic variation in AG are still unknown.

The current study aimed to comprehensively analyze the inter-sex genomic variation, as well as inter-species variation, between AG and *A. ruthenus* (AR, sterlet), a related tetraploid sturgeon species. By mapping the AG reads to the AR genome, we validated the sex-specificity in AG of the female-specific region previously described in AR. We quantified the SNP and short indel-based variation between the species (AG and AR) as well as between the sexes in AG. We identified 15 peaks of sex-associated variation (10 kb segments with 30 or more sex-associated variants in each segment), 1 of which matched the previously reported sex-variable region. Finally, we identified 14 known and predicted genes in proximity to these peaks.

## 2. Results

### 2.1. Sequencing and Assembly of AG Draft Genome

#### 2.1.1. Assembly Metrics

We assembled the genome of the Russian sturgeon (AG) from paired-end WGS reads from five females and five males. The assembly statistics can be found in Appendix A. The draft genome size is 2,389,722,506 bp arranged into 3,393,603 contigs, with N50 = 1043 bp. The assembly and raw data were deposited in Genbank (Acc. No. JANDEF000000000) and SRA (BioProject Accession: PRJNA851573), respectively.

We then used BUSCO with the Actinopterygii dataset of 3640 orthologs to assess the completeness of the assembled genome. Of the 3640 genes in the dataset, 780 genes (21.4%) were completely retrieved in the assembled genome, including 714 (19.6%) single-copy genes and 66 (1.8%) duplicated genes. In addition, 555 (15.2%) genes were fragmented, and 2305 (63.4%) genes were not found. Thus, the BUSCO assembly completeness estimate was 21.4 (Appendix A).

#### 2.1.2. Mapping and Coverage Depth Statistics

We aligned the reads from the 10 individual libraries to the AG genome. Total mapping rates were very high for all libraries (above 99.6%, Appendix A). According to samtools statistics, about 82–84% of the reads were properly paired, while 15–17% of the reads fell into the category “with mate mapped to a different chr”, probably due to the small size of contigs in the assembled genome.

We also measured the genome coverage and the average depth in the individual libraries, the aggregation of sex-specific libraries, and all libraries together. The genome coverage ranged from 92.2 to 93.3%, and the average depth ranged from 17.4 to 22.9 reads for the individual 10 samples. No significant differences were observed between females and males (Figure 1b,c). The aggregated coverage for all fish was 99.65%, and the average depth was 165.8 reads (Appendix A).

### 2.2. Alignment to a Related Species Genome—A. ruthenus (AR)

We next repeated the mapping and coverage analysis using the published AR male and female genomes as a reference (GCF_010645085.1 and GCA_902713425.1, respectively). Total mapping rates were very high for all libraries against both genome versions (above 97.3%, Appendix A). The proportion of mapped reads that were properly paired was about 89% (compared to 82–84% in mapping to the assembled genome), presumably due to a better assembly level of these reference genomes.

For the AR male genome reference, the genome coverage ranged from 94.9 to 95.3%, and the average depth ranged from 22.9 to 30.2 reads for the individual 10 samples. No significant differences were observed for female vs. male libraries. The aggregated coverage for all fish was 97.5%, and the average depth was 228.6 reads (Figure 2a,b and Appendix A). For the female genome reference, which was shorter by almost 200 Mb, the genome coverage ranged between 98.1 and 98.4%, slightly above the coverage observed for the male genome as a reference, and the average depth ranged from 25.7 to 33.9 reads for the individual 10 samples. As with the male reference, no significant differences were observed for female vs. male libraries. The aggregated coverage for all fish was 99.7%, and the average depth was 250 reads (Figure 2c,d and Appendix A). On the genomic scale, no significant differences were observed between the male and female AR genome as a reference, nor between mapping and coverage results of libraries from different sexes.

### 2.3. Quantifying Inter-Species Variation (between AG and AR)

To evaluate the variation between the AG and AR species, we used a pipeline for variation discovery (see Methods). We performed our analysis in two configurations, considering the variation against male or female AR genomes. For each configuration, we identified homozygous variants (present in the WGS reads of all 10 AG samples with an allele frequency (AF) of 1, regardless of sex).

A total of 2,802,343 and 2,855,785 variants were identified when considering female and male reference genomes, respectively (Appendix A). The average distance between inter-species variations was 662.7 bp, and the average number of inter-species variations per 10 kb segment was 3.7, when mapping to the AR male genome (Figure 3a,b); mapping to the AR female resulted in an average distance of 582.6 and 4.9 variant positions on 10 kb segments on average (Figure 3c,d). When excluding segments with no variant positions, the average variations per segment were 16.1 and 17.8 for the AR male and AR female, respectively. Of the identified variants, SNPs accounted for ~90.2 and 87.5% of the variants for female and male reference genomes, respectively, while indels accounted for 9.8 and 12.5% of the variants, respectively (Figure 3e,f).

### 2.4. Substitution Rates in Individual Genes Identified in the AG Assembly

We used BUSCO-predicted genes in the AG assembly to estimate the ratio of the nonsynonymous codon substitution rate to that for synonymous codons (dN/dS). We started the analysis with 714 complete and single-copy BUSCO genes. We chose to perform the analysis in relation to AR, the closest species in evolutionary terms to a complete reference. For 713 genes, an AR homologue was found. For 657 of the genes, dS ranged from 0.01 to 2; the rest of the genes were filtered out, as dS values outside of this range could result in unreliable dN/dS estimation. dN/dS ranged from 0.001 to 4.6067 (Figure 3g, Appendix A). dN/dS ratios larger than 1 indicate adaptive selection while dN/dS ratios smaller than 1 indicate purifying selection. A total of 643 of the genes had a dN/dS < 1, while only 14 had dN/Ds > 1, suggesting a strong bias towards purifying (conservative) selection pressure on most genes identified in AG. A notable exception was *pdrg1* (p53 and DNA-damage-regulated gene 1), with a dN/dS value of 4.6, indicating a strong adaptive selection.

### 2.5. Quantifying Inter-Sex Variation in AG

First, we screened the mapping results of the five male and five female libraries against the AG genome, looking for regions with (1) no coverage in female samples and coverage of at least 10 reads in at least three male samples and (2) vice versa. We found 315,614 positions across 3075 contigs with zero coverage in females, and 96,775 positions across 1694 contigs for the opposite case. However, the high degree of fragmentation of the AG assembly led us to use a more comprehensive genome assembly from a related species (AR) as a reference genome.

Therefore, we evaluated the variation between sexes in AG using a pipeline for variation discovery (see Methods), considering the variation against the female AR genome. We chose the female genome as a reference as it is known to contain at least one sex-specific fragment. We identified 7382 variant positions in which all AG males were homozygous to the variation (with AF = 1, 1/1 genotype) while all AG females either had a heterozygous variant (0/1) or no variant (0/0) in the same position. The average distance between inter-species variations was 203,260 bp, and the average number of variations per 10k segment was close to zero (Figure 4a,b). When excluding segments with no variant positions, the average number of variations per segment was 2.3. Of the identified variants, SNPs accounted for ~85.8%, while indels accounted for 14.2% of the variants (Figure 4c).

### 2.6. Identification of Clusters of Inter-Sex Variation in AG

We scanned the AR female genome with segments of 10k bp length. Altogether, we identified 3151 segments with at least one variant position. The segments belonged to 92 out of 245 contigs (Figure 5a). A total of 23 of the segments had more than 25 variations, while 15 of them had 30 or more variations. The segment with the highest number of variations at 88 was found on contig CACTIG010000152.1. The previously described sex-variable region on contig CACTIG010000179.1 is associated with a cluster of 32 inter-sex variants (Figure 5a, highlighted).

### 2.7. Identification of Genes Nearest to Clusters of Inter-Sex Variation

Using gene annotation from the AR male genome, we mapped gene regions on the AR female genome. Then, for each sex-associated variant position, we found its closest gene in the AR female genome (Appendix A). A total of 3723 out of 7382 variant positions fell inside genes (distance 0 in Appendix A). Table 1 shows the genes nearest to the top 15 sex-variable variant clusters (with 30 or more variable positions). Functional enrichment analysis of this short gene list, using g:Profiler with the zebrafish (*D. rerio*) database, identified the GO-MF term GO:0030021 (extracellular matrix structural constituent conferring compression resistance) as significantly enriched (P_adj_ = 0.0499). No KEGG pathways or other categories represented in g:Profiler showed significant enrichment.

## 3. Discussion

Although sturgeon genomes have been extensively studied [17], to our knowledge, this is the first study to present a quantitative analysis of inter-sex genomic variation in the sturgeon, and to compare it with whole-genome inter-species variation within the sturgeon family.

To assess inter-sex and inter-species variation, we generated whole-genome sequencing from five AG males and five females. We performed the first attempt to assemble the AG genome using the sequenced libraries. The resulting assembly was highly fragmented and had low completeness estimation. Nevertheless, we were able to find regions that were covered exclusively by female or male libraries. More accurate assembly of the AG genome will be advantageous not only for studying sex determination mechanisms but also for other evolutionary studies. This goal can be achieved by incorporating longer read sequencing and mate-pair sequencing [18], or by using improved genome assembly approaches, such as reference-guided assembly [19]. Combining AG genomic and transcriptomic data (e.g., our previously published gonadal transcriptomes [10]) will enhance the identification and characterization of protein-coding regions and a better understanding of AG evolution and biology.

Due to the mentioned limitations of the AG assembly and the small sample size, we used the genome of a closely related species, *A. ruthenus* (AR), as a reference and developed a stringent approach to identify genomic variants. In particular, we performed the identification of variants in each individual sample and filtered low-quality variants. For inter-species variation, we reported variants that were identified in all 10 fish. For inter-sex variation, we reported homozygous variation in all five male fish that was either heterozygous or missing in all five females. In the latter analysis, low-quality variations in females were also included to reduce the discovery of false-positive sex-associated variations. In addition, we required that the reported positions have certain sequencing coverage in all sequenced libraries.

The genetic mechanisms that control sex determination have been studied extensively in vertebrates. In contrast with the prevalence of sex-specific chromosomes in mammals, a variety of genetic and environmental mechanisms have been described in fish [20,21] and amphibians [22,23]. This complexity stems both from the plasticity of sex determination in individual species, and the variability in sex determination methods between the groups [24]. Several genes involved in sex determination and differentiation have been described in teleost fishes, including, notably, the male-specific *Amhy*, *Dmrt1*, *Dmy*, *Gsdf*, *SdY*, and *Sox3*, and the female-specific *Foxl_2_* and *Foxl_3_* (reviewed in [25]). Nevertheless, the genetic mechanisms of sex determination remain elusive in most fish species, including a well-studied model organism such as zebrafish (*D. rerio*), where a polygenic sex determination system was suggested [26] or a species of prime economical significance such as salmon (*S. salar*), where genomic instability has reportedly resulted in a high mobility of the sex-determining genomic region [27]. Both polygenic sex determination and genomic instability, evidenced by successive genomic duplications, are also features of the AG genome and agree with our findings. From the evolutionary standpoint, sex determination mechanisms in the Acipenseridae (as “living fossils”) may predate those of most other fish, and are therefore of particular interest. Even though the inter-sex variation is conserved in the Acipenseridae, and thus much more ancient than the speciation events within it, the inter-sex variation in AG on all chromosomes/contigs is much smaller than the inter-species variation between AG and AR. This is in line with the notion that genetic sex determination in the Russian sturgeon is not based on sex chromosomes [5], but rather on one or several short sex-variable genomic fragments, putatively affecting the expression of one or more functional genes.

Our results in AG agree with the findings of Kuhl et al. [14] and subsequent studies that successfully used the AllWsex2 marker to identify the sex of different sturgeon species [14,15,16]. Furthermore, we identified 15 additional genomic regions in AR with clusters of 30 or more inter-sex variants in the AG WGS reads that map to them. Follow-up studies are necessary to validate these sex-variable regions in additional samples and assess their evolutionary conservation, evolutionary history, and potential for aquacultural application and species conservation/repopulation efforts.

Despite considerable recent progress in sex identification in sturgeons in the context of research and aquaculture, the molecular mechanisms controlling sex determination and sexual differentiation in this family remain obscure. Our study identified 15 genes in close proximity to clusters of >30 inter-sex variants (within 10 kb genomic fragments), including the region identified by Kuhl et al. [14]. Of particular interest among these is orofacial cleft 1 candidate gene 1 protein homolog (*ofcc1*) and the homologous LOC117394522, the genes nearest to the previously described sex-variable region. Although the genes are present and are expressed in both sexes (data not shown), a cluster of sex-associated SNPs in and near the beginning of this gene points towards possible functional involvement. These, as well as the other identified genes, may have functional roles in the physiological processes of sex determination and sexual differentiation in the sturgeon. All the genes on our list need to be validated in a larger sample before they are considered viable candidates, whose putative roles should be explored in future studies. Additionally, the planned incorporation of long-read sequencing data will help improve the contiguity of the AG genome assembly.

## 4. Materials and Methods

The overall scheme of our study is presented in Figure 1a.

### 4.1. Fish and Sampling Procedure

Samples were taken from the dorsal fins of four-year-old male and female Russian sturgeon (4–6 kg) from the Caviar Galilee fish farm in Israel. The fish were anesthetized with 0.03% tricaine methane sulfonate (MS222, Sigma-Aldrich, St. Louis, MO, USA), and their length and body weights were quantified [10]. Pieces (about 2 cm) from the dorsal fins were sampled, and gonad biopsies were obtained by endoscopy from males and females, as described in detail [4] for sex identification by microscope. Fin samples were stored at −80 °C until further analysis.

### 4.2. DNA Extraction

DNA was extracted from fin samples using a Sigma GenElute Genomic DNA kit (G1N70-1KT), with the heat block sample homogenization step shortened to 10 min to minimize degradation of DNA. The concentration and purity of the DNA were measured using a NanoDrop 8000 Spectrophotometer from Thermo-Fisher Scientific (Waltham, MA USA) and a Bioanalyzer 2100, Agilent (Santa Clara, CA, USA).

### 4.3. Genome Sequencing

A total of 10 samples (5 female, 5 male) were shipped on dry ice to the Beijing Genomics Institute (BGI, Hong Kong), where libraries were prepared and sequenced on BGISEQ-500 equipment. After removing adaptor sequences and low-quality reads, between 467 and 627 million clean 100-base paired-end reads were obtained from each sample. The Q20 quality score was above 97.3% for all samples.

### 4.4. De Novo Genome Assembly

De novo AG genome assembly was performed based on the paired-end reads from all 10 fish, using the MEGAHIT assembler v1.2.9 [28]. We refer to this genome as the “AG genome” throughout the manuscript. We applied Benchmarking Universal Single-copy Orthologs (BUSCO) 5.2.1 [29,30] for assessing genome completeness, by using the “actinopterygii_obd10” lineage dataset containing a set of 3640 core genes, genome mode, and Augustus as a gene predictor (-l, -m, and -f options, respectively).

### 4.5. Additional Reference Genomes

We downloaded two additional genomes from NCBI for the closely related species *A. ruthenus* (AR): male (GCF_010645085.1) and female (GCA_902713425.1). We refer to these genomes as the “AR male” and “AR female” genomes throughout the manuscript.

### 4.6. Estimation of Nucleotide Substitution Rates (dN/dS)

We retrieved single-copy protein sequences that were identified by BUSCO from the AG genome. We then used BLAST against GCF_010645085.1_ASM1064508v1_translated_cds.faa to retrieve their orthologues in the AR male genome. For each protein, we generated two FASTA files, one for amino acid sequences (faa) and one for nucleic acid sequences (fna); both files contained AG and AR orthologues. The dN/dS was calculated as described [31]. Briefly, we aligned the faa file with clustal omega [32]. We applied pal2nal [33] with the parameter “-nogap” to obtain a codon-based nucleic acid alignment. We then applied the CODEML function implemented in PAML [34] to calculate dN, dS, and dN/dS. We used a script provided by [31] to filter genes for which dS was <0.01 or >2, as dS values lower than 0.01 indicate that the sequences are too similar, and dS values above 2 indicate that the sequences are too divergent, which in both cases could result in unreliable dN/dS estimates.

### 4.7. Genome Mapping and Coverage Assessment

For each of the above genomes, we used BWA (v.0.7.17-r1188) [35] to index the genome and to align each individual library read to the genome using BWA-MEM with standard parameters. We converted the alignment SAM files to BAM files using samtools view v1.12 (with -S -b parameters). We sorted BAM files by coordinates using samtools sort. We generated the mapping statistics with samtools flagstat.

We created a depth file for each library using samtools depth (with parameters -aa -f list of all BAM files). With an in-house python script, we computed depth frequencies and depth statistics histograms for each sample, for the aggregation of sex-specific samples, and for all samples together.

### 4.8. Identifying Contigs Covered by Reads from Only One of the Sexes

We searched for contigs in the AG genome that were covered by only one of the sexes using the depth files. To that end, using an awk command, we extracted all positions, for which all five female samples had a depth of 0 and at least three of the male samples had a depth of 10, and vice versa. We then counted the number of unique contigs that contained the identified positions.

### 4.9. Variant Discovery

We performed variant calling on each of the 10 samples aligned to either (1) the AR male genome or (2) AR female genome using BCFtools v1.13 [36]. First, BCF files were created using bcftools mpileup. Second, VCF files were created from BCF files using bcftools call (with parameters -m -v). We denoted these files as “Unfiltered_VCF”. To reduce false-positive variant calls, we processed each VCF file and kept only records with QUAL>30 using BCFtools view v1.13 (with -i “QUAL>30”). We denoted these files as “Filtered_VCF)”. The unfiltered and filtered files were indexed using the BCFtools index.

To identify variation between AG and AR, we merged filtered VCF files of females and males aligned against (1) the AR male genome and (2) AR female genome, with BCFtools merge. Using an in-house python script, we extracted from each merged file all positions in which homozygous variation was present in all 10 AG samples (AF = AC/AN = 1, equivalent to 1/1 genotype).

To identify sex-associated variation in AG, we considered VCF files produced from alignment to the AR female genome. We merged male filtered VCF files and female unfiltered VCF files with BCFtools merge. Using an in-house python script, we extracted from the merged file all positions that complied with the following two criteria: (a) AG males that had a homozygous (AF = 1, 1/1 genotype) variant compared to the AR female reference; and (b) AG females that had either a heterozygous variant (0/1) or no variant (0/0) in the same position. In order to make sure that SNPs were not identified in the female genome due to lack of coverage, the merged file with the above positions was intersected with the depth file using Bedtools intersect (with parameters -wb -wa). In-house script extracted positions with a coverage of >2 in all samples.

We applied bcftools stats to obtain the statistics for each variation group. With in-house python scripts, we calculated and plotted the number of SNPs per segment (10 kb) and the distance between SNPs.

### 4.10. Identification of Genes Adjacent to Sex-Associated Variants

We extracted from the *features table* of the AR male genome, downloaded from NCBI (GCF_010645085.1_ASM1064508v1_feature_table.txt.gz), genomic coordinates of all mRNAs. Then, we extracted the DNA sequences that corresponded to these coordinates into a fasta file. Using minimap2 (V2.24-r1122) [37], we aligned the fasta file against AR female genomes. We processed the output file to keep for each query one hit with the highest coverage of the query, and then converted it to a BED file. We applied BEDtools v2.30.0 closest (with parameters -d -k 1 -t “first”) to find the sex-associated variant positions’ closest gene in the AR female genome.

### 4.11. Functional Enrichment Analysis

We used g:Profiler [38] (https://biit.cs.ut.ee/gprofiler/gost, accessed on 9 July 2022) to check for GO terms and KEGG pathways enriched in the gene list obtained in 4.9. *D. rerio* (zebrafish) database, with the default parameters.

## Figures and Tables

**Figure 1 ijms-23-09469-f001:**
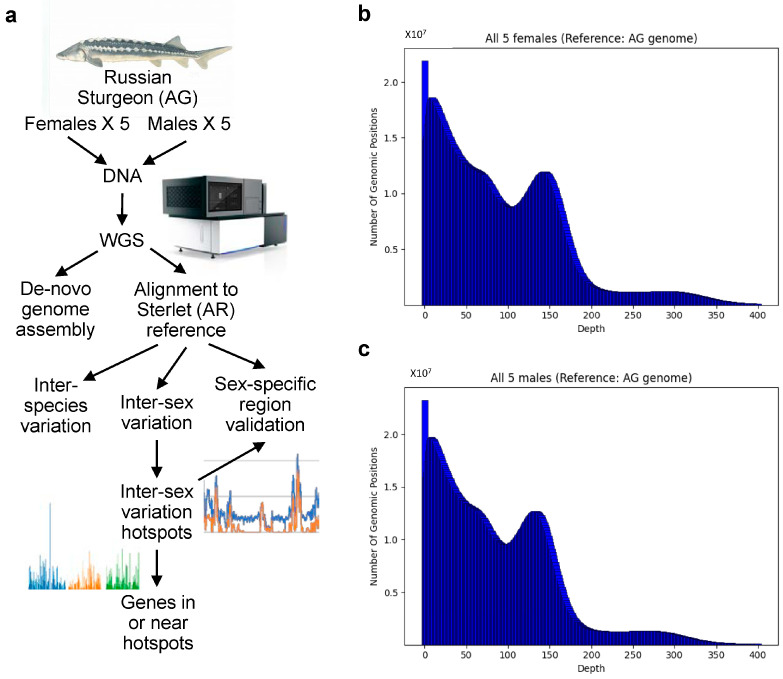
(**a**) Scheme of study; (**b**,**c**) Distribution of sequencing depth over all genome positions when reads are aligned to AG genome; (**b**) female samples combined and (**c**) male samples combined. Distributions for individual samples and for all samples combined can be found in Appendix A.

**Figure 2 ijms-23-09469-f002:**
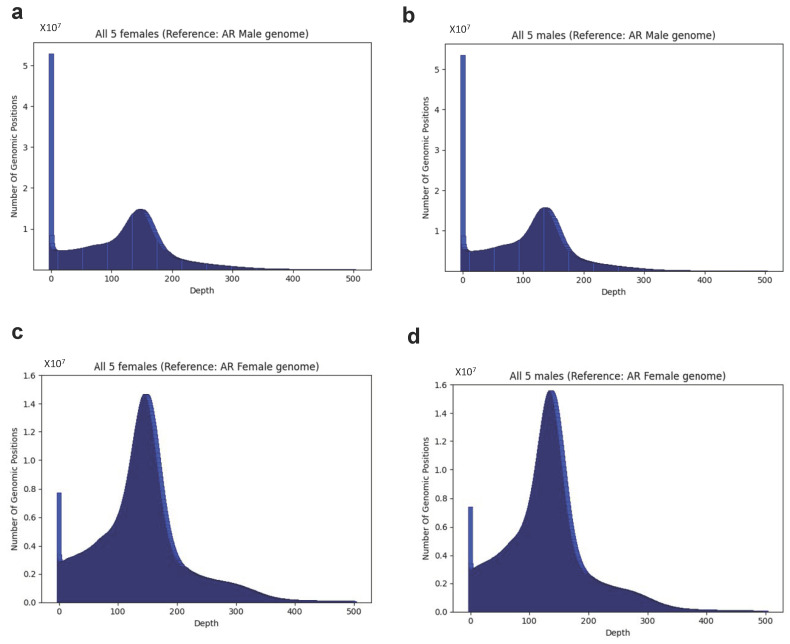
(**a**–**d**) Distribution of sequencing depth over all genome positions when AG reads are aligned to the AR male or female reference genome: (**a**) female AG reads aligned to male AR reference; (**b**) male AG reads aligned to male AR reference; (**c**) female AG reads aligned to female AR reference; (**d**) male AG reads aligned to female AR reference. Distributions for individual samples and for all samples combined can be found in Appendix A.

**Figure 3 ijms-23-09469-f003:**
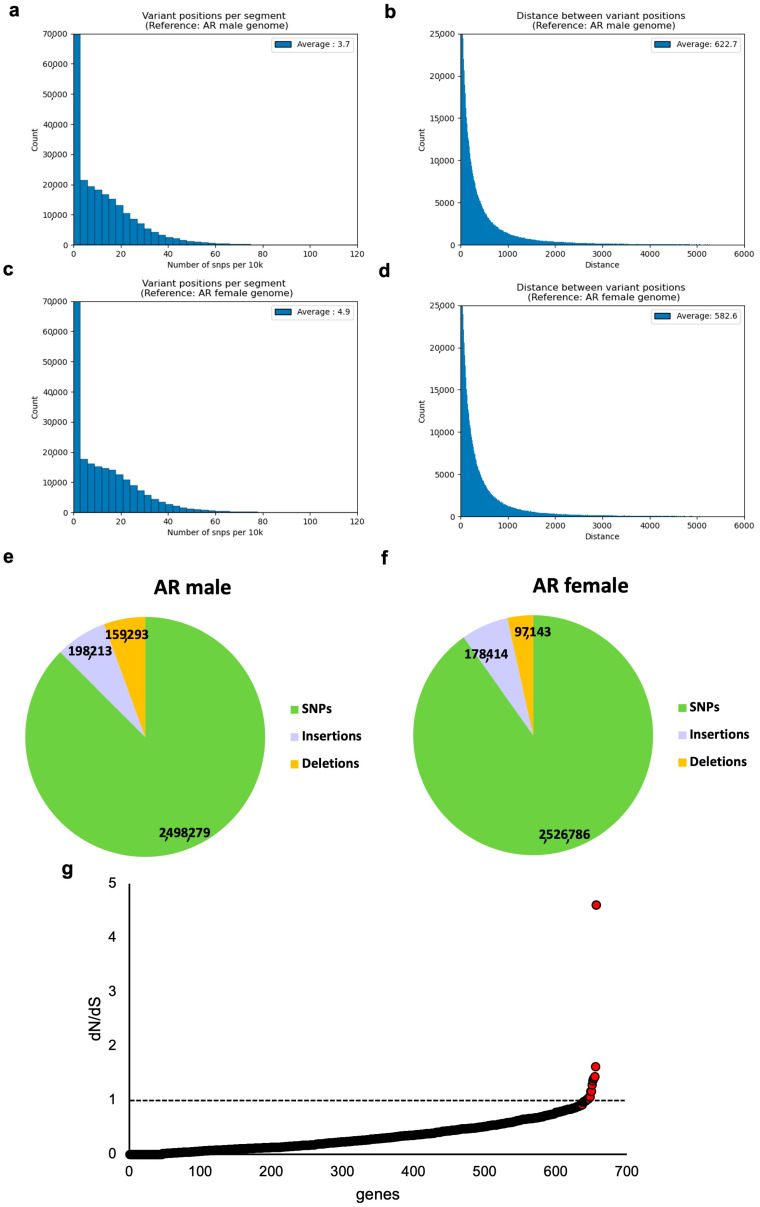
(**a**–**d**) Distributions of variant positions representing inter-species genomic variation between AG and AR, when mapping to AR male or AR female reference genome: (**a**) variant positions per 10 kb segment, AR male; (**b**) distance between variant positions, AR male; (**c**) variant positions per 10 kb segment, AR female; (**d**) distance between variant positions, AR female. In (**a**,**c**), average considers all segments, including segments with no variations. (**e**,**f**) Pie chart representing the contribution of different types of variants (SNPs, insertions, and deletions) to the overall variation observed when mapped to (**e**) male and (**f**) female AR genomes. (**g**) dN/dS nucleotide substitution ratios for 657 genes identified in the AG assembly, relative to AR. Note the outlier gene, *pdrg1*. Dashed line indicates dN/dS = 1, or neutral selection.

**Figure 4 ijms-23-09469-f004:**
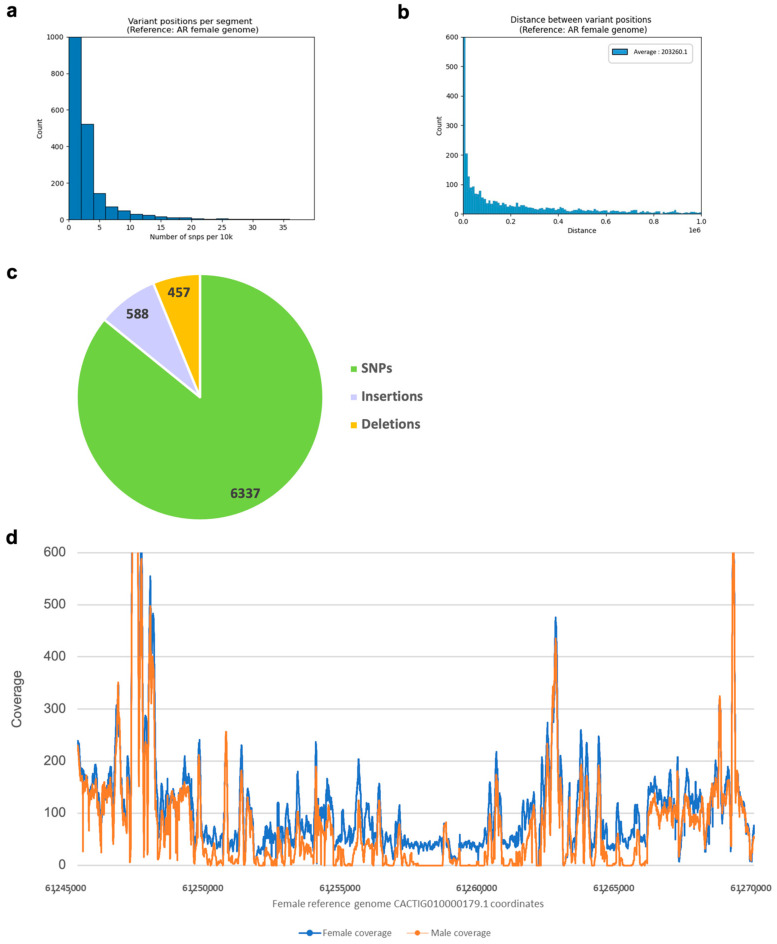
Inter-sex variation in AG. (**a**,**b**) Distributions of SNPs representing inter-sex genomic variation in AG, when mapping to the female reference. (**a**) SNPs per 10 kb segment; (**b**) distance between SNPs. (**c**) Pie chart representing the contribution of different types of variants (SNPs, insertions, and deletions) to the overall variation observed. (**d**) Aggregated coverage of AG WGS reads from 5 females (blue) and 5 males (red) in previously reported sex-variable region of the AR genome (female reference, contig CACTIG010000179.1). Note female-specific fragments between the positions 61,250,000 and 61,267,000 of the reference contig.

**Figure 5 ijms-23-09469-f005:**
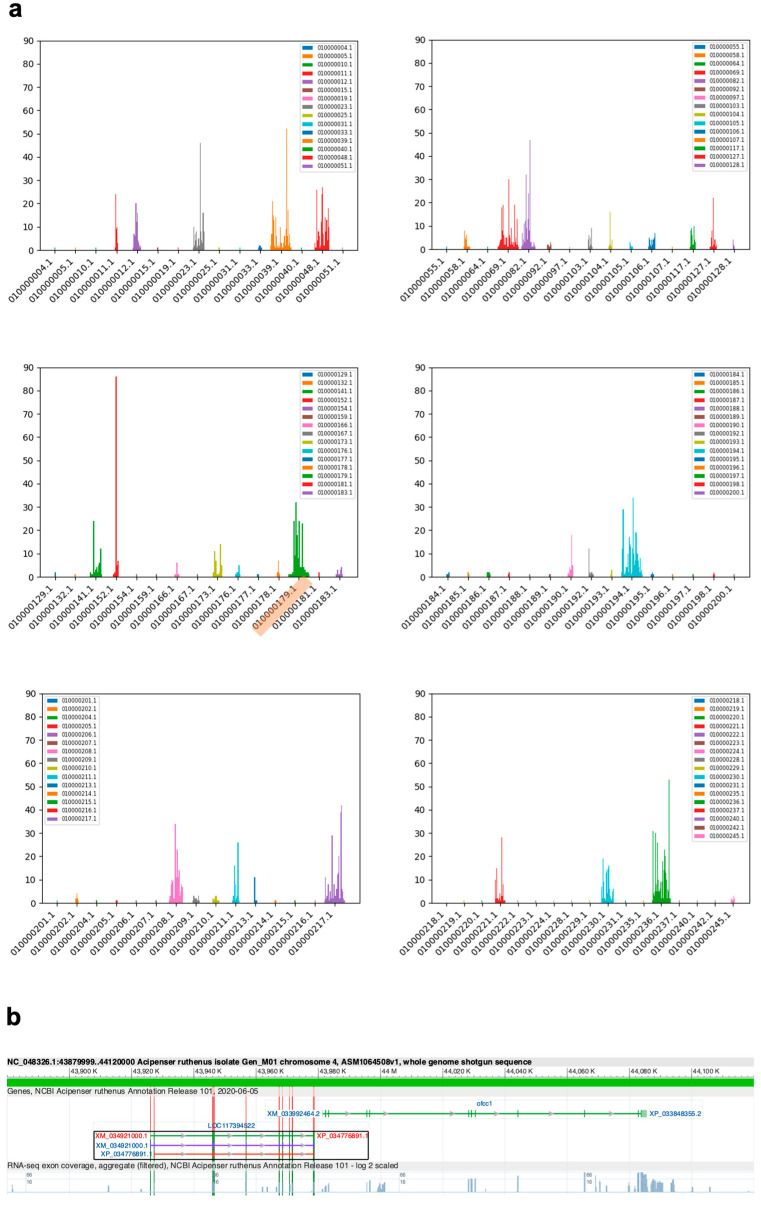
(**a**) Counts of inter-sex variation in AG in each 10 kb fragment, mapped to contigs in the AR female reference genome. Only segments with at least one variation, across 92 out of 245 contigs, are shown. The previously described sex-variable region maps to contig CACTIG010000179.1 (highlighted) and is associated with a cluster of 32 inter-sex variants. The first five panels show 15 contigs, and the last one shows 17 contigs. (**b**) Genome browser view of ofcc1 and the homologous LOC117394522, the genes nearest to the previously described sex-variable region, mapped to the AR male reference genome.

**Table 1 ijms-23-09469-t001:** Known and predicted genes nearest to the top 15 sex-variable variant clusters identified using AG WGS reads and AR female reference genome. *ofcc1*, the gene nearest to the previously reported sex-variable region, is in bold.

Reference Sequence	Start *	End *	Variants	Nearest Gene	Short Name	In Gene?
CACTIG010000023.1	35,214,339	35,219,825	46	NC_048332.1|vascular_cell_adhesion_molecule_1b	*vcam1b*	+
CACTIG010000039.1	31,510,109	31,517,637	52	NC_048333.1|sorting_nexin-7-like	*snx7*	−
CACTIG010000069.1	54,920,692	54,925,300	30	NC_048323.1|sorting_nexin-25-like	*snx25*	+
CACTIG010000082.1	11,280,103	11,289,542	32	NC_048336.1|decorin-like	*dcn*	−
CACTIG010000082.1	27,990,061	27,998,319	47	NC_048336.1|dnaJ_homolog_subfamily_B_member_9-like	*dnajb9*	−
CACTIG010000152.1	22,143,064	22,149,608	86	NC_048348.1|potassium_voltage-gated_channel_subfamily_C_member_1	*kcnc1*	+
**CACTIG010000179.1**	**61,245,978**	**61,249,200**	**32**	**NC_048326.1|orofacial_cleft_1_candidate_gene_1_protein_homolog**	** *ofcc1* **	**+**
CACTIG010000194.1	91,630,461	91,639,996	34	NC_048325.1|neuropilin_and_tolloid-like_protein_1	*neto1*	−
CACTIG010000208.1	13,330,045	13,338,213	34	NC_048342.1|neuronal_migration_protein_doublecortin-like	*dcx*	−
CACTIG010000217.1	58,035,319	58,039,159	39	NC_048327.1|phosphatase_and_actin_regulator_2-like	*phactr2*	+
CACTIG010000217.1	58,052,280	58,058,468	42	NC_048327.1|phosphatase_and_actin_regulator_2-like	*phactr2*	+
CACTIG010000236.1	7,300,133	7,309,147	31	NC_048329.1|protein_mono-ADP-ribosyltransferase_PARP12-like	*parp12*	+
CACTIG010000236.1	11,730,268	11,739,819	37	NC_048329.1|SH3_and_multiple_ankyrin_repeat_domains_3a	*shank3a*	+
CACTIG010000236.1	12,145,816	12,147,030	30	NC_048329.1|protein_FAM107B-like	*fam107b*	−
CACTIG010000236.1	45,581,476	45,589,775	53	NC_048329.1|synaptic_vesicular_amine_transporter-like	*slc18*	+

* Start and end indicate the position of the first and last variant positions in the segment, respectively.

## Data Availability

The assembly and raw data were deposited in Genbank (Acc. No. JANDEF000000000) and SRA (BioProject Accession: PRJNA851573), respectively.

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
