# Peer review of "Whole-Genome Inter-Sex Variation in Russian Sturgeon (Acipenser gueldenstaedtii)"

_ijms, 2022, doi:10.3390/ijms23169469_

Round 1
Reviewer 1 Report
"Whole genome inter-sex variation in Russian sturgeon" by Degani, Sarel, Hajouj, Hurvitz, Veksler-Lublinsky, and Meerson presents preliminary data towards identifying sex determination mechanisms in Russian sturgeon. The study and the manuscript are very incomplete.
The genomes that are assembled in this study are very incomplete with most of the gene level data missing. It is probably better for this data to be published and in the literature than not, so I don't have a major problem with the incompleteness of the data for that reason, although what you can say/ do with this data is obviously very limited.
From the limited data that are generated, the authors generate a set of genes that may be sex-specific given a pipeline and set of assumptions (eg heterozygosity in females consistent with expectations). However, that analysis is just presented as an essentially raw result. At a bare minimum, perhaps the limited data that are analyzed will generate some patterns in clustering in pathways in KEGG that would enable more process-based hypotheses to be generated. Other downstream analyses that don't fall apart because of missing data should also be performed.
Can anything be done with dN/dS using the 843 complete genes, even if this doesn't illuminate just sex determination mechanisms?
Given error rates and the limited sample sizes, what is the expected false positive rate? Other statistical calculations are also missing.
Lastly, the discussion is very incomplete. There should be a richer discussion of possible sex determination mechanisms (drawing from literature knowledge of mechanisms that are known in other fish species as well as vertebrates more broadly), what the genetics that will result from this could be, and how it relates to both the pipeline that was implemented and the results that were obtained.
Despite the incompleteness of everything, I am not fully negative towards eventual publication of this study. Still, more work needs to be done to make both the study and the writing less minimalistic.
Author Response
Dear Drs.,
We thank you for considering our manuscript (ijms-1811437) titled “Whole-genome inter-sex variation in Russian sturgeon (Acipenser gueldenstaedtii)” for publication, and we thank the reviewers for their helpful comments and suggestions. Following these, we have extensively revised our manuscript, and we believe that these changes have substantially improved it. A point-by-point response follows, with the reviewers’ comments italicized.
Reviewer 1:
The genomes that are assembled in this study are very incomplete with most of the gene level data missing. It is probably better for this data to be published and in the literature than not, so I don't have a major problem with the incompleteness of the data for that reason, although what you can say/ do with this data is obviously very limited.
We added two paragraphs in the Discussion (starting at line 236) to expound on the limitations of the AG assembly, and to emphasize the different strategies that we used to overcome these limitations in our downstream analysis (namely, alignment to a more complete assembly from a related species - AR, and stringent filtering of results).
From the limited data that are generated, the authors generate a set of genes that may be sex-specific given a pipeline and set of assumptions (eg heterozygosity in females consistent with expectations). However, that analysis is just presented as an essentially raw result. At a bare minimum, perhaps the limited data that are analyzed will generate some patterns in clustering in pathways in KEGG that would enable more process-based hypotheses to be generated. Other downstream analyses that don't fall apart because of missing data should also be performed.
We thank the reviewer for this suggestion. We used g:Profiler to check for GO terms enriched in the gene list, and identified the GO-MF term GO:0030021 (extracellular matrix structural constituent conferring compression resistance) as significantly enriched (Padj=0.0499).In general, we cannot expect pathway analysis to be particularly helpful in this case, as only a small number of genes likely take active part in sex determination, and furthermore all the genes on our list need to be validated in a larger sample before they are considered viable candidates. The Results (starting at line 219), Discussion (starting at line 294), and Methods (starting at line 319) were amended accordingly.
Can anything be done with dN/dS using the 843 complete genes, even if this doesn't illuminate just sex determination mechanisms?
This is indeed an interesting direction to further characterize the genome of AG. However, to perform the dN/dS analysis, the coding region within the identified genes must be determined/predicted. This requires integration with transcriptome data and will be addressed in a planned follow-up study. We have added a reference to this future direction in the Discussion (paragraph starting at line 236).
Given error rates and the limited sample sizes, what is the expected false positive rate? Other statistical calculations are also missing.
The reviewer is correct in pointing out that the sample size is small, which limits the use of a statistical significance test (used in GWAS studies) to identify sex-associated SNPs. To overcome this, we used a stringent approach to identify genomic variants. This included identification of variants in each individual sample, filtering low quality variants, and reporting variants that were identified in all 10 fish or all 5 fish of the same sex, for inter-species variation or inter-sex variation in AG, respectively, with sequencing coverage above the threshold in the reported areas. We emphasized this point in the Discussion (paragraph starting at line 248).
Lastly, the discussion is very incomplete. There should be a richer discussion of possible sex determination mechanisms (drawing from literature knowledge of mechanisms that are known in other fish species as well as vertebrates more broadly), what the genetics that will result from this could be, and how it relates to both the pipeline that was implemented and the results that were obtained.
We thank the reviewer for this suggestion. In agreement, we have expanded the Discussion to address the complexity of possible sex determination mechanisms in fish and to quote several comprehensive reviews, as well as recent studies on this topic (paragraph starting at line 258).
Despite the incompleteness of everything, I am not fully negative towards eventual publication of this study. Still, more work needs to be done to make both the study and the writing less minimalistic.
We thank the reviewer for the comprehensive assessment and the valuable comments, and hope that our revised manuscript addresses them in a satisfactory fashion, while our future efforts stand to address those aspects beyond its current scope.
In addition to changes requested by the reviewers, we have corrected several minor errors and inconsistencies in the text, and an accession number was added for the raw data. (The accession number for the assembly is still pending and will be added before publication.) We would like to thank both reviewers and the Editorial Office and hope that our revised manuscript merits publication in the International Journal of Molecular Sciences.
Sincerely,
On behalf of the authors:
Dr. Isana Veksler-Lublinsky and Dr. Ari Meerson
Reviewer 2 Report
Minor editorial comments are as follows:
1. line 18: add commas before and after therefore
2. lines 24, 57, 65, 103, 266: Acipenser should be A.
3. lines 70, 147, 148, 167, 172, 179, 187, 226 and throughtout the text: 10kb should be 10 kb
4. line 95: 92.2% - 93.3% should be 92.2 - 93.3%
5. line 110: 94.9% - 95.3% should be 94.9 - 95.3%
6. line 115: 98.1% - 98.4% should be 98.1 - 98.4%
7. line 135: add a comma after genomes
8. lines 136, 138, 309: 10k should be 10 k
9. line 141: 90.2% and 87.5% should be 90.2 and 87.5%
10. line 216: add [14] after Kuhl et al.
11. line 240: add St. Louis, MO, USA after Sigma-Aldrich,
12. line 249: add (city, state, country) after Scientific
13. line 250: add (Agilent, city, state, country) after Bioanalyzer
14. line 273: move a period after (...parameters)
15. line 276: move a period after (... bam files)
16. All references: titles should be small capitals
17. scientific names of animals should be italic: lines 346, 353, 362, 365, 373, 376, 385,
18. journal names should be spelled out: lines 360, 373, 387
Author Response
Dear Drs.,
We thank you for considering our manuscript (ijms-1811437) titled “Whole-genome inter-sex variation in Russian sturgeon (Acipenser gueldenstaedtii)” for publication, and we thank the reviewers for their helpful comments and suggestions. Following these, we have extensively revised our manuscript, and we believe that these changes have substantially improved it. A point-by-point response follows, with the reviewers’ comments italicized.
Reviewer 2:
Minor editorial comments are as follows:
- line 18: add commas before and after therefore
- lines 24, 57, 65, 103, 266: Acipenser should be A.
- lines 70, 147, 148, 167, 172, 179, 187, 226 and throughtout the text: 10kb should be 10 kb
- line 95: 92.2% - 93.3% should be 92.2 - 93.3%
- line 110: 94.9% - 95.3% should be 94.9 - 95.3%
- line 115: 98.1% - 98.4% should be 98.1 - 98.4%
- line 135: add a comma after genomes
- lines 136, 138, 309: 10k should be 10 k
- line 141: 90.2% and 87.5% should be 90.2 and 87.5%
- line 216: add [14] after Kuhl et al.
- line 240: add St. Louis, MO, USA after Sigma-Aldrich,
- line 249: add (city, state, country) after Scientific
- line 250: add (Agilent, city, state, country) after Bioanalyzer
- line 273: move a period after (...parameters)
- line 276: move a period after (... bam files)
- All references: titles should be small capitals
- scientific names of animals should be italic: lines 346, 353, 362, 365, 373, 376, 385,
- journal names should be spelled out: lines 360, 373, 387
We thank the reviewer for the favorable assessment and the minor editorial comments. All of these were implemented in the revised manuscript, except items 17-18, as the reference style for our manuscript was automatically formatted by a reference management software (Zotero) in accordance with the MDPI default style.
In addition to changes requested by the reviewers, we have corrected several minor errors and inconsistencies in the text, and an accession number was added for the raw data. (The accession number for the assembly is still pending and will be added before publication.) We would like to thank both reviewers and the Editorial Office and hope that our revised manuscript merits publication in the International Journal of Molecular Sciences.
Sincerely,
On behalf of the authors:
Dr. Isana Veksler-Lublinsky and Dr. Ari Meerson
Round 2
Reviewer 1 Report
The authors have included a brief survey of sex determination mechanisms in fish in the discussion, which is an improvement over the previous version. They have included a GO enrichment analysis, but not one from KEGG, which might give more pathway-level specificity to their findings. They have not done any dN/dS or other downstream analyses that are common in genome analysis papers. They have not done a full statistic analysis to validate findings either.
Author Response
Dear Drs.,
We thank you for considering our manuscript (ijms-1811437) titled “Whole-genome inter-sex variation in Russian sturgeon (Acipenser gueldenstaedtii)” for publication. Following the Reviewer’s comments, we have further revised our manuscript. A point-by-point response follows, with the Reviewer’s comments italicized.
Reviewer 1:
The authors have included a brief survey of sex determination mechanisms in fish in the discussion, which is an improvement over the previous version.
We thank the Reviewer for the acknowledgement.
They have included a GO enrichment analysis, but not one from KEGG, which might give more pathway-level specificity to their findings.
We used g:Profiler (https://biit.cs.ut.ee/gprofiler/gost) to check for both GO terms and KEGG pathways. However, no KEGG pathways showed significant enrichment among the gene list, which is not surprising given the small number of genes in the list (15). We have emphasized this in the Results (lines 249-250) and Methods (lines 435-436) sections in the revised manuscript.
They have not done any dN/dS or other downstream analyses that are common in genome analysis papers.
Following the Reviewer’s suggestion, we have performed a dN/dS substitution rate analysis on 657 complete and single-copy BUSCO-predicted AG genes, using the AR genome as a reference. We made the relevant additions to the Results (item 2.4., lines 173-185), added a new panel (g) to Fig. 3, and described the method in the Methods section (item 4.6., lines 363-378).
They have not done a full statistic analysis to validate findings either.
We are not certain what is being referred to by the reviewer. A literature search yields many published articles with analysis similar to ours, with no further statistical analysis of the genomic variation data. Examples:
- Kuhl H, Guiguen Y, Höhne C et al. A 180 Myr-old female-specific genome region in sturgeon reveals the oldest known vertebrate sex determining system with undifferentiated sex chromosomes. Trans. R. Soc. (2021). http://doi.org/10.1098/rstb.2020.0089
- Nam BH, Yoo D, Kim YO et al.Whole genome sequencing reveals the impact of recent artificial selection on red sea bream reared in fish farms. Sci Rep 9, 6487 (2019). https://doi.org/10.1038/s41598-019-42988-z
- Santos CA, Andrade SCS, Freitas PD. Identification of SNPs potentially related to immune responses and growth performance in Litopenaeus vannamei by RNA-seq analyses. PeerJ6:e5154(2018). https://doi.org/10.7717/peerj.5154
In addition to changes requested by the Reviewer, we have corrected several minor errors and inconsistencies in the text, and an accession number was added for the assembly as well as the raw data. We would like to thank again the Reviewers and the Editorial Office and hope that our revised manuscript merits publication in the International Journal of Molecular Sciences.
Sincerely,
On behalf of the authors:
Dr. Isana Veksler-Lublinsky and Dr. Ari Meerson
Round 3
Reviewer 1 Report
The authors have probably done as much as is reasonable to ask at this stage for the journal it is being published in. I am ok to see it published as is at this point. Please do make sure that all methods described are properly referenced, including programs used, and that enough information is included to make sure that the study can be replicated. For example, how was dN/dS calculated?
Author Response
Following the Reviewer’s comments, we have further revised our manuscript. A point-by-point response follows, with the Reviewer’s comments italicized.
Reviewer 1:
The authors have probably done as much as is reasonable to ask at this stage for the journal it is being published in. I am ok to see it published as is at this point.
We thank the Reviewer for the acknowledgement.
Please do make sure that all methods described are properly referenced, including programs used, and that enough information is included to make sure that the study can be replicated. For example, how was dN/dS calculated?
We made sure that all methods used are referenced. The dN/dS calculation is described in the Methods section, item 4.6 (lines 330-342). Additional minor corrections are highlighted in yellow, in the revised version of the manuscript.
We would like to thank again the Reviewers and the Editorial Office and hope that our revised manuscript merits publication in the International Journal of Molecular Sciences.